# Relationship Between Urinary Copper, Zinc, and Cadmium and Kidney Damage Biomarkers in Young People

**DOI:** 10.3390/ijms26167980

**Published:** 2025-08-18

**Authors:** Manolo Ortega-Romero, Elodia Rojas Lima, Olivier C. Barbier, Octavio Gamaliel Aztatzi-Aguilar, Juan Carlos Rubio-Gutiérrez, Juana Narváez Morales, Mariela Esparza García, Ángel Barrera-Hernández, Mónica I. Jiménez-Córdova, Luz María Del Razo, Pablo Mendez-Hernández, Mara Medeiros

**Affiliations:** 1Unidad de Investigación en Salud en el Trabajo, Centro Médico Nacional “Siglo XXI”, Instituto Mexicano del Seguro Social, Ciudad de Mexico 06720, Mexico; rom_0@hotmail.com (M.O.-R.); elodia.rojas.lima@gmail.com (E.R.L.); 2Departamento de Toxicología, CINVESTAV IPN, Ciudad de Mexico 07360, Mexico; obarbier@cinvestav.mx (O.C.B.); oaztatzi@cinvestav.mx (O.G.A.-A.); ldelrazo@cinvestav.mx (L.M.D.R.); 3Unidad de Investigación en Nefrología y Metabolismo Mineral Óseo, Hospital Infantil de México Federico Gómez, Ciudad de Mexico 06720, Mexico; 4Facultad de Ciencias de la Salud, Universidad Autónoma de Tlaxcala, Tlaxcala 90750, Mexico; pmendezh@hotmail.com; 5Departamento de Farmacología, Facultad de Medicina, National Autonomous University of Mexico, Ciudad de Mexico 04510, Mexico

**Keywords:** low levels of cadmium, copper–zinc ratio, biomarkers of early kidney damage, pediatric population

## Abstract

Chronic kidney disease (CKD) is a global public health issue linked to toxic elements like cadmium (Cd) and mercury (Hg), which harm the kidneys even at low exposure levels. Copper (Cu) and zinc (Zn) imbalances could exacerbate inflammation, oxidative stress, and kidney damage because the Cu/Zn ratio could be a critical marker of renal dysfunction. The study evaluated 914 adolescents aged 11–18 through urine samples to assess the presence of kidney damage biomarkers (OPN, KIM-1, CLU, NGAL, and Cys-C) by using Luminex Magpix and trace metals (Cd, Hg, Cu, Zn) by using ICP-mass. Overweight (18.71%) and obesity (12.58%) rates were noted. Cd and Cu showed positive correlations with kidney damage biomarkers, while Zn exhibited protective effects. Regression models indicated that Cd exposure increased kidney damage markers, emphasizing the importance of Cu/Zn ratio. Environmental exposure to Cd affects kidney health even at low levels, as the Cu/Zn ratio correlates with kidney damage markers in low-Cd exposure, suggesting that the Cu/Zn ratio could participate in the nephrotoxicity process, highlighting trace element imbalance as a potential predictor of kidney function decline.

## 1. Introduction

Chronic kidney disease (CKD) is one of the world’s most important public health problems; hence, characteristics such as dietary nutrition and research on toxic and/or trace elements have attracted considerable interest [1]. Heavy metals, such as cadmium (Cd) and mercury (Hg), are widely distributed environmental toxicants that pose significant health risks, including kidney disease. Exposure to these metals has been associated with an increased risk of CKD, which may occur even in element levels previously considered safe, suggesting that their nephrotoxic potential has been underestimated [2].

In Mexico, studies have reported on environmental exposure to potentially toxic elements and their impact on kidney health [3,4]. Regions with elevated Cd exposure have been associated with kidney dysfunction. Additionally, studies conducted on Mexican children and adolescents have reported that low levels of exposure to Cd and Pb correlate with kidney alterations, emphasizing the vulnerability of young populations to metal-induced nephropathy and highlighting the complex interactions between multiple environmental contaminants and kidney function [4,5,6].

On the other hand, exposure to Cd or Hg causes the dysregulation of essential elements involved in homeostasis, such as zinc (Zn), copper (Cu), iron (Fe), and calcium (Ca) [7]. In particular, Cu and Zn are essential micronutrients that seem to predict the development and course of a few diseases, for they are related to inflammatory parameters and oxidative stress processes [8]. Their insufficiency, deficiency, or toxic levels can increase the incidence of degenerative conditions such as vascular disease, cancer, infections, and kidney disease [9].

Zn is an important mineral and a constituent of more than 300 enzymes involved in key physiological processes, including anti-inflammatory and antioxidant (copper/zinc superoxide dismutase) functions, immune responses, apoptosis, and transcription regulation (zinc finger) [10]. In childhood, Zn deficiency or metabolic disorders related to Zn have been associated with mood disorders, attention-deficit hyperactivity disorder (ADHD), and tic disorders [11]. Zn deficiency is also frequently associated with infections, cancer, atherosclerosis, and other age-related degenerative diseases. In the kidney, it aggravates fibrosis by promoting inflammatory processes [1]. Several studies have reported that Cd affects Zn homeostasis, with lower Zn serum levels observed in subjects with high Cd body burden [12]. Additionally, autopsy studies have shown that Zn levels in the liver and kidney cortex decreased with age, often correlated with Cd levels [13]. Cd-induced renal tubular damage has been attributed to oxidative stress mechanisms [12].

Meanwhile, Cu is an important contributor to bone health, the antioxidant system, and as a cofactor in diverse enzymatic processes. However, excessive Cu can result in oxidative stress and the overproduction of free radicals [14], which in turn can trigger inflammatory processes and increase the risk of cancer development [10]. In renal toxicity, oxidative stress is a key mechanism of Cu to induce damage, as lipoperoxidation alters cellular membrane integrity, leading to renal dysfunction [15]. When Zn and Cu levels become imbalanced, characterized by low Zn and high Cu, oxidative stress increases, impairing multiple enzyme functions [10]. An elevated ratio between Cu and Zn (Cu/Zn-R) above 2.0 has been associated with an inflammatory response or a decreased nutritional Zn status, particularly in the elderly population [8].

These findings suggest that the Cu/Zn-R may serve as a better indicator of various diseases compared with Zn or Cu levels alone [15]. High serum Cu/Zn-R has been found in people with debilitating conditions, cancer malignancy, and all-cause mortality in older subjects, including observational cohort evidence of an increased risk of cardiovascular diseases, heart failure, and cancer, as well as infectious diseases such as pneumonia and kidney diseases [8,9]. Similar to Cu, it has been reported that an imbalance in the Cu/Zn-R has been associated with renal dysfunction in humans, which is mediated by oxidative stress [7].

Given the documented impact of heavy metal exposure on kidney health in Mexico, it is essential to identify populations at risk and explore additional markers associated with kidney dysfunction. The Cu/Zn-R has emerged as a potential indicator of kidney damage. This study aims to evaluate the impact of the Zn/Cu-R on kidney damage biomarkers in young individuals exposed to low-level Cd, considering its potential role in kidney health.

## 2. Results

A total of 914 patients were evaluated; 55.03% were women. The population’s median age was 13 years and ranged between 12 and 18 years. To determine the nutritional state of the participants, we considered parameters like body mass index (BMI), calculated with the methodology of the CDC for the pediatric population, and waist-to-height ratio (WtHr); BMI classification presented that 18.71% of the participants were overweight and 12.58% were obese; with the WtHr, we observed a median value of 0.47, which is less than 0.5, a value that is considered normal. Concerning kidney function, we observed a median eGFR of 103.75 and an albumin–creatinine ratio (ACR) of 5.08 mg/g urinary creatinine (UCr). The median concentrations of other substances included NGAL (5.31 ng/mL), KIM-1 (0.25 ng/mL), a1-MG (27.94 mg/mL), OPN (59.78 ng/mL), Cys-C (1.2 ng/mL), and Clu (132.02 ng/mL). Finally, we observed the median concentrations of Cd (0.013 ng/mL), Hg (0.013 ng/mL), Zn (617.06 ng/mL), and Cu (24.73 ng/mL), all in urine; the median of the population’s Cu/Zn-R was 0.044 (Table 1).

In Table 2, we present correlations of Cd, Hg Cu, and Zn with the biomarkers of early kidney damage, including eGFR and ACR. Cd showed positive and significative correlations with NGAL, KIM-1, OPN, and Cys-C, indicating that the higher the concentration of Cd was, the higher the concentrations of the biomarkers of early kidney damage were; the same behavior was observed between Cu and NGAL, KIM-1, a1-MG, OPN, Cys-C, and CLU; a negative and significative correlation was observed with eGFR; Hg only showed correlation with CLU; and Zn presented significant negative correlations with NGAL only.

We used regression models with explanatory variables to explore the potential effects of Cd on the urinary excretion levels of Hg, Zn, Cu, and Cu/Zn. Our observations revealed that Cd in the continuous scale was associated with an increase in log-Hg (β): 0.159, 95% CI: 0.089; 0.228). On the other hand, levels over the limit of detection (≥0.069 ng/mL) were associated with an increase in the log-Zinc (β: 0.169, 95% CI: 0.050; 0.287), an increase in the log-Hg (ß: 0.981, 95% CI: 0.509; 1.453), and a decrease in the log-Cu/Zinc ratio (β:−0.237, 95% CI: −0.352; −0.123) in comparison to the <LOD category (Table 3).

In the linear regression models adjusted for confounders (Table 4), we observed that each unit of increase in log-Cd showed a positive and significant association with log-OPN (β: 0.079, 95% CI: 0.035; 0.123) and log-Cys-C (β: 0.064, 95% CI: 0.001; 0.126) in the continuous scale; when Cd was dichotomous (≥0.069 vs. <LOD), we observed an increase in the urinary levels of log-OPN (β: 0.431, 95% CI: 0.151; 0.712) and a decrease in log-CLU (ß: −0.259, 95% CI: −0.513; −0.005). Each unit of increase in log-Hg only showed a positive and significant association with log-OPN (β: 0.05, 95% CI: 0.008; 0.092) in the continuous scale; when Hg was dichotomous (≥0.079 vs. <LOD), we observed an increase in the urinary levels of log-OPN (β: 0.531, 95% CI: 0.254; 0.809) and an increase in log-Cys-C (β: 0.522, 95% CI: 0.147; 0.897).

In Table 4, the log-Cu/Zn-R presents a positive and significative association with all biomarkers of early kidney damage, a negative association with eGFR (β: −0.018, 95% CI: −0.002; −0.033), and an increased risk of presenting albuminuria (OR_ACR≥30 vs. ACR<30_: 1.42, 95% CI: 1.16; 1.74). Also, with the increase of the log-Cu/Zn-R, we observed an increase in the biomarkers of early kidney damage levels, suggesting present renal damage (Figure 1). 

## 3. Discussion

Environmental exposure to xenobiotics, such as cadmium (Cd) and mercury (Hg), is a public health risk due to their bio-accumulative and toxic effects, which can alter renal function as nephrons are exposed [16]. In this work, the median concentration of Cd and Hg in urine was 0.013 ng/mL, which is considered safe and low as compared to the reference values. The biomonitoring equivalent (BE) is 1.5 ng/mL for Cd [17] and 35 mcg/g-Cr for Hg [18]. In this study, we could not consider the presence of Cd and Hg in urine to be such a public health problem; for this to be possible, the percentage of individuals exposed to these elements should have been higher or the concentrations found should have exceeded the values considered as maximum permissible limits. However, adverse effects, including nephrotoxicity, can occur even at these low concentrations [19,20]. Our results try to show that the presence of toxic elements such as Cd can cause alterations in the kidney when these toxic elements are in the body at the same time that we observe anomalies in essential elements for the body, indicating that the interactions or synergies between the elements can cause effects that could be deleterious to the body.

Cd can accumulate in the renal cortex and contribute to tubular and glomerular damage [21]. The median concentration of Cd in this population was 0.013 ng/mL, a very low level compared to other studies. However, these levels should not be considered insignificant, as the nephrotoxic effects of Cd at low levels remain controversial. In the Korean adult population, the concentration range was 0.95–1.36 ng/mgCr [22]; in healthy children aged 13–18 years in Henan, China, the median concentration was 0.64 ng/mL (0.31, 1.00) [23].

Low Cd levels have been positively correlated with N-acetyl-β-D-glucosaminidase (NAG) activity and urinary β2-microglobulin (β2-MG), but negatively correlated with eGFR [22]. In healthy kidney donors aged 24–70 years, renal Cd concentrations of 13 ng/mg induced signs of tubular atrophy, glomerulosclerosis, and/or arteriosclerosis [24]. We observed a relationship between the presence of low levels of Cd and the expression of NGAL, KIM-1, OPN, and Cys-C (in adjusted models OPN and Cys-C only), suggesting possible renal damage induced by low levels of Cd. Finally, in animal models, low Cd induces the presence of NAG, alkaline phosphatase (ALP), β2-MG, and KIM-1 [25].

On the other hand, the kidney is considered the primary site of accumulation and intoxication of mercury [26]; unfortunately, the available information on nephrotoxicity is sparse. In a population with hypertension aged 8–17 years, Hg concentrations of 37 ng/mL were observed [27], while artisanal miners in Ghana had concentrations of 10 ng/mL [28], Colombia reported 9.9 ng/mL [29], and Brazil reported 17.3 ng/g creatinine with a range of 0.1 to 301 ng/mL [30]. As with Cd in this report, the median concentration of Hg was very low (0.013 ng/mL) compared to other studies.

Hg did not show clear relationships with markers of kidney injury, but when we categorized Hg (undetectable Hg and ≥0.079 ng/mL), we observed that individuals with detectable Hg were significantly associated with OPN and Cys-C. In artisanal mercury mining workers from Mexico, the Hg concentration was 503.4 mg/mgCr, presenting associations with BUN, urea, and OPN [3], the last biomarker similar to our study. In animal studies, the excretion of CLU, HMGB1, KIM-1, MCP-1, and β2-microglobulin increased significantly in a mercury-dependent manner [31].

The metabolic balance of trace elements should be considered when assessing renal health risks from toxic metal exposure, as they play an important role in maintaining health [32]. Both Zn and Cu are among these trace elements with important biological functions, and their deficiency or excess can cause various problems [11]. When trace elements are altered, health problems caused by toxic metals (including Cd, Hg, Pb, etc.) may be exacerbated.

Zn is an essential nutrient with the biological equivalent (BE) for deficiency ranging from 159 to 206 ng/mL, while for protection against toxicity, the range is 439–3489 μg/L [33]; in this work, the median concentration observed was 617 ng/mL, a concentration in the protective range. Cu is not routinely assessed in clinical practice, so no reliable reference values have been identified [34], but there are values in populations. In a cohort of healthy adolescents from the Czech Republic, the mean concentration was 14.68 ng/mL [35]; in firefighters during wildland firefighting operations, urinary Cu levels were 63.5 ng/mL [36]; and 25 ng/mL Cu was the value for participants aged from 6 to 19 years from the Canadian Health Measures Survey 2007–2013 [37], the latter of which contains very similar values to those we observed (24.73 ng/mL).

The relationship between Zn and renal function remains controversial; it has been attributed to a protective role against oxidative damage, inflammation, and fibrosis (mainly in diabetic populations), and it has also been suggested that Zn may slow renal damage [38]. A cross-sectional study of 816 Japanese subjects failed to show an independent association between zinc and eGFR [1]. In China, 461 elderly subjects showed a protective effect of zinc against CKD only in older adults aged 90 years and older [39]. We observed a negative correlation of Zn with NGAL (the only biomarker with a significant association), suggesting a protective role.

Increased urinary Cu has been shown to cause direct damage to proximal tubule epithelial cells [1]. It can also lead to oxidative damage, glomerular endothelial atherosclerosis, and damage to the renal filtration barrier [40]. In a Chinese population, increased odds of CKD were associated with increasing concentrations of Cu (OR: 1.90, 95% CI = 1.59, 2.29) [41]. In animal studies, Cu was observed to bioaccumulate in the renal cortex and was associated with oxidative damage and increased 4-HNE, KIM-1, and NGAL [42]. We presented positive correlations between all biomarkers of early renal injury and Cu, as well as a negative correlation with eGFR, suggesting that high levels of Cu also cause renal injury in this population.

Thus, the disruption of trace elements in the kidney may be a mechanism underlying the nephrotoxicity of exposure to Cd and Hg. Disturbed Zn and Cu homeostasis may serve as a predictor of the deterioration of renal function due to exposure to Cd and Hg. The ratio of Cu and Zn (Cu/Zn-R) was associated with the low-level Cd group concerning the Cd-undetectable group. Curiously, no relationship was observed between the Cu/Zn-R and Hg, but if there is a relationship between Cd and Hg, that suggests that there exist interactions with other metals and metalloids in the body. This result can be explained by the simultaneous exposure to Cd, Hg, Cu, and Zn, elements found in the environment and food. A study of copper smelter workers reported positive correlations between Cd, Cu, and Zn in tissues [43]; chronic low-level Cd exposure was associated with reduced Cu and Zn reabsorption [44].

A previous study showed that the Cu/Zn-R was a sensitive predictor of various causes of mortality in the population [15]. Unfortunately, most epidemiological studies have been carried out in serum. In a Thai study, Cd exposure was associated with increased serum Cu/Zn-R [44]; in a study of 299 healthy Croatian men aged 20–55 years, Cd exposure was associated with decreased serum Zn levels [45]; and in an Australian autopsy study, an effect of Cd on Zn and Cu homeostasis in the liver and kidneys was observed. However, the observed results correlate with our findings.

We observed positive and significant associations of Cu/Zn-R with all biomarkers of early renal damage, including ACR and decreased eGFR. Previous studies have reported that the balance of essential metals, such as Cu and Zn, plays an important role in renal dysfunction, especially in groups with low Cd exposure [46]. For example, in the low Cd exposure group, urinary β2-MG concentration was positively associated with Cu/Zn-R, suggesting that a Cu/Zn-R imbalance may be an independent factor for elevated urinary biomarker levels [7]. Although these results suggest that Cu/Zn-R has a different mechanism of renal toxicity than Cd, they also suggest that Cd-induced kidney damage can be controlled by balancing essential metals.

The propensity of zinc to compete/inhibit Cd reabsorption may explain the findings of both our study and other studies showing an increased risk of renal Cd toxicity in those with low zinc status [47]. Cd-induced copper and zinc dysregulation may contribute to its toxicity and the pathogenesis of Cd-related diseases (hypertension, diabetes, and macular degeneration) [48,49].

In conclusion, Cu/Zn-R showed associations with biomarkers of renal damage in a population exposed to very low levels of Cd. Cd levels were also associated with lower Cu/Zn-R. Thus, Cu/Zn-R may serve as a predictor of kidney function deterioration in populations exposed to low levels of Cd, and it needs to be evaluated in order to identify kidney damage caused by exposure to other toxic metals, as well as to evaluate the role of Zn and Cu as a strategy to prevent or mitigate the toxicity of various xenobiotics and their adverse health effects. Finally, we cannot exclude the possibility that exposure to other environmental toxicants may have confounded the observed results. In addition, the participants’ eGFR, ACR, and biomarkers of early kidney damage were based on a single spot measurement. The lack of information on the diet of participants may affect the form in which trace elements are excreted in urine.

## 4. Materials and Methods

### 4.1. Study Design

The cross-sectional study was conducted on 914 apparently healthy individuals of both sexes aged from 11 to 18 years that were recruited from 2019 to September 2023. They were recruited from Tlaxcala, Chiautempan, and Apetatitlán, three municipalities from the state of Tlaxcala, because health professionals and local authorities reported an increase of 300% in hospitalizations for kidney disease of uncertain etiology in under-25-year-olds from 2004 to 2012. Additionally, a 2016 pilot study in Apizaco, Tlaxcala, presented a 4.3% prevalence of CKD in the population of 6- to 15-year-olds [50]. Exclusion criteria included the following: a history of renal disease, the presence of fever 48 h before the study, physical activity for >2 h the day before sampling, and females who were on their menses. The sample size was calculated considering a 95% confidence interval (CI), a precision of 5%, and an expected prevalence of 0.31% (Tlaxcala State Registry of CKD cases, 2016). The study was conducted with a sample of individuals from Tlaxcala between 11 and 18 years of age and comprising both sexes. The study was conducted following the Declaration of Helsinki and was approved by the ethics committee of the Hospital Infantil de México Federico Gómez (HIMFG) HIM 2019/025 SSA 1242, by the Institutional Bioethics Committee for Research in Humans (COBISH-Cinvestav) CINVESTAV 034/2016, and by the ethics committee of the Hospital Infantil de Tlaxcala: HIT/CEI/2021/03. Having signed the letters of assent and informed consent, the participants also attended the medical review and the socioenvironmental questionnaire. Urine was obtained during the first morning, and 1.5 mL aliquots were made; blood samples were obtained by peripheral venipuncture and stored at −70 °C.

### 4.2. General Evaluation

The participants underwent a medical review and a socio-environmental questionnaire that included household characteristics, chronic-degenerative diseases, personal and pathological history, environmental exposure, etc. Blood pressure was measured with a sphygmomanometer, weight was determined with a clinical scale, and height was determined with a stadiometer. BMI was calculated with the STAT Growth Charts TM application version 3.2 from the Center for Disease Control (CDC).

### 4.3. Biochemical Analysis

The biochemical assessment was performed with the Clinitek Status Plus device (Siemens Healthcare Global). UCr was evaluated by the Jaffe reaction (Randox Laboratories Ltd., Crumlin, UK), and the albumin–creatinine ratio (ACR) was calculated with the RANDOX MICROALBUMIN (mALB)-immunoturbidimetric assay for albumin kit. The serum creatinine (SCr) was processed in the central reference laboratory at HIM using DIMENSION RxL Max SIEMENS equipment. The estimated glomerular filtration rate (eGFR) was estimated with the Bedside–Schwartz formula, traceable by non-isotope dilution mass spectrometry [51].

### 4.4. Biomarkers of Early Kidney Damage Assay

The determination of early renal damage biomarkers: Urinary levels of NGAL, Cys-C, OPN, clusterin (CLU), alpha-1-microglobulin (α-1MG), and KIM-1 were determined by Luminex xMAP^®^ technology (Millipore Corp., Burlington, MA, USA) using a R&D System Magnetic Luminex^®^ Performance Assay Human Kidney Biomarker Base Kit following the product instructions. All samples were analyzed in duplicate. The analysis indicated coefficients of variation between 6% and 14% and a precision of 80–120% depending on the molecule studied.

### 4.5. Determination of Cadmium, Mercury, Copper, and Zinc in Urine

The concentration of urinary Cd, Hg, Cu, and Zn was measured using an ICP-Mass Perkin Elmer ICP-MS, NexION 300D according to the procedure described by the Laboratory of Research and Service in Toxicology (LISTO) of Cinvestav, accredited by the Mexican accreditation entity (EMA) (INV-0007-013/19). The samples were nebulized and entered the plasma generated by the argon gas. The ions formed in the plasma were introduced into the mass analyzer (quadruples). They were classified according to their mass–charge ratio and directed to the simultaneous dual detector, which generated a signal proportional to the concentration of the element. The elements were quantified using a validated method based on a calibration graph, evaluating duplicate samples for the elements, including blank samples and at least six different concentrations. Certified reference samples of trace elements in urine with different concentrations, acquired from the INSPQ (INSPQ/toxicologie QM-U-Q1705, urine), were used to validate the analysis of the concentrations of elements in urine. The results were expressed in ng/mL. The study indicated coefficients of variation less than 10% and a precision of 80–120%. The limits of detection (LOD) were 0.069 ng/mL for Cd, 0.079 ng/mL for Hg, 0.112 ng/mL for Cu, and 0.368 ng/mL for Zn.

### 4.6. Statistical Analysis

The study population was described using proportions for categorical variables and median and interquartile range for numerical variables. The biomarkers of early kidney damage and metals found below the limit of detection were imputed with the iterative Markov Chains Monte Carlo (MCMC). The dilution effect in metals and renal parameters was corrected for urinary density (by Levine-Fahy) [52].

Spearman correlations were used to evaluate the relationship between metals and early kidney damage biomarkers, albuminuria, and glomerular filtration rate. Due to the asymmetry of the data, the exposure and outcome variables were log-transformed. The association between metals and renal parameters was evaluated using linear regression models adjusted for confounders selected by the conceptual framework.

The influence of Cd urinary levels in Zn, Cu, Hg and Cu/Zn-R excretion was explored. Because 91.8% of Cd and 94.3% of Hg determination in urine samples were below the detection limit, a dichotomous variable was used where 0 represented this category, and 1 represented the subjects with urinary Cd and Hg detection. Then, linear regression models compared the levels of Zn, Cu, and Cu/Zn excretion in the Cd and Hg categories using age, sex, BMI, and poverty as potential confounders. *p*-values < 0.05 were considered statistically significant. The statistical analysis was performed with Stata Software version 18.

## Figures and Tables

**Figure 1 ijms-26-07980-f001:**
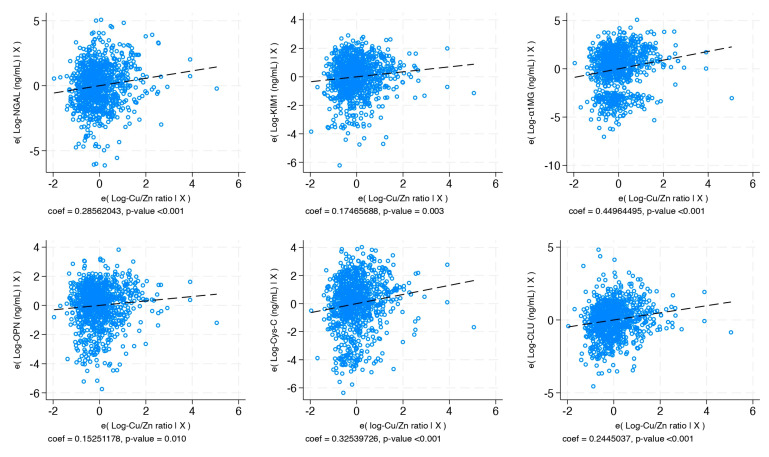
Relationship between urinary Cu/Zn ratio and early kidney biomarkers (NGAL, KIM-1, CLU, Cys-C, OPN, and α-1MG). Models were adjusted for confounders (age, sex, BMI, and poverty).

**Table 1 ijms-26-07980-t001:** Study population characteristics.

Characteristic	*n* = 914
*Age(years), median (IQR)*	13 (12; 15)
*Sex, n (%)* Female Male	503 (55.03) 411 (44.97)
*BMI, n (%)* Underweight Normal Overweight Obese	20 (2.19) 608 (66.52) 171 (18.71) 115 (12.58)
*WHtR, median (IQR)*	0.47 (0.44; 0.52)
*Smoking status*	
Nonsmoker	879 (96.17)
Former smoker	22 (2.41)
Current smoker	13 (1.42)
*Poverty, n (%)* Yes No	525 (57.44) 389 (42.56)
*Renal parameters, median (IQR)*	
eGFR (ml/min por 1.73 m^2^)	103.75 (92.04; 121.52)
ACR (mg/g-creatinine)	4.94 (<LOD; 22.53)
*Early kidney biomarker (ng/mL) ^a^ median (IQR)*	
NGAL	5.31 (1.59; 14.92)
KIM-1	0.25 (0.11; 0.52)
α-1MG OPN	27.94 (5.53; 79.66) 59.78 (23.45; 126.68)
Cys-C	1.20 (0.44; 3.31)
CLU	132.03 (62.86; 264.99)
*Urinary Metals (ng/mL) ^a^ median (IQR)*	
Cadmium	0.013 (0.004; 0.036)
Mercury	0.013 (0.003; 0.043)
Zinc	617.06 (413.99; 818.01)
Copper	24.73 (17.90; 39.07)
Copper/Zinc ratio	0.044 (0.029; 0.070)

Abbreviations: IQR, interquartile range; BMI: body mass index (underweight: <percentile 5, normal weight: percentile 5 to percentile 85, overweight: percentile 85 to percentile 95, obese: >95); WHtR, waist-to-height ratio; eGFR, estimated glomerular filtration rate; ACR, albumin–creatinine ratio; NGAL, neutrophil gelatinase-associated Lipocalin; KIM-1, Kidney Injury Molecule 1; α-1MG, α-1-microglobulin; OPN, Osteopontin; Cys-C, Cystatin C; CLU, Clusterin; LOD, limit of detection. ^a^ Early kidney damage biomarkers and metals were adjusted for urinary density and expressed as ng/mL.

**Table 2 ijms-26-07980-t002:** Correlation between renal parameters and metals.

Description	NGAL Rho	KIM-1 Rho	α-1MG Rho	OPN Rho	Cys-C Rho	CLU Rho	eGFR Rho
Cadmium	0.082 *	0.085 *	0.042	0.093 *	0.065 *	0.015	0.027
Mercury	0.044	0.023	0.042	0.060	0.053	0.069 *	0.014
Zinc	−0.097 *	0.001	−0.034	0.050	−0.058	0.001	−0.046
Copper	0.117 **	0.104 *	0.154 **	0.106 *	0.105 *	0.175 **	−0.074 *
Copper/Zinc Ratio	0.169 **	0.095 *	0.173 **	0.060	0.148 **	0.163 **	−0.019

Abbreviations: eGFR, estimated glomerular filtration rate; NGAL, neutrophil gelatinase-associated Lipocalin; KIM-1, Kidney Injury Molecule 1; α-1MG, α-1-microglobulin; OPN, Osteopontin; Cys-C, Cystatin C; CLU, Clusterin. Early kidney damage biomarkers and metals were adjusted for urinary density and expressed as ng/mL. rho, Spearman correlation. * *p* < 0.05, ** *p* < 0.001.

**Table 3 ijms-26-07980-t003:** Urinary metal excretion according to urinary cadmium levels.

Description	Log-Zinc β (95% CI)	Log-Copper β (95% CI)	Log-Mercury β (95% CI)	Log-Cu/Zn-R β (95% CI)
Cadmium *n* = 914	0.017 (−0.002; 0.035)	0.017 (−0.009; 0.043)	0.159 (0.089; 0.228)	0.0003 (−0.026; 0.026)
<LOD *n* = 839	Ref.	Ref.	Ref.	Ref.
≥0.069 *n* = 75	0.169 (0.050; 0.287)	−0.069 (−0.198; 0.061)	0.981 (0.509; 1.453)	−0.237 (−0.352; −0.123)

Abbreviations: LOD, limit of detection; Cu/Zn-R, Copper/Zinc Ratio. Metals were adjusted for urinary density and expressed as ng/mL. Models adjusted for age (years), sex (female/male), body mass index (normal/underweight/overweight/obese), poverty (yes/no).

**Table 4 ijms-26-07980-t004:** Association among metals and early kidney damage biomarkers, estimated glomerular filtration rate, and albuminuria (*n* = 914).

Description	Log-NGAL β (95% CI)	Log-KIM-1 β (95% CI)	Log-α-1MG β (95% CI)	Log-OPN β (95% CI)	Log-Cys-C β (95% CI)	Log-CLU β (95% CI)	Log-eGFR β (95% CI)	ACR (≥30 vs. <30) OR (95% CI)
Log-Cadmium *n* = 914	0.035 (−0.022; 0.092)	0.044 (−0.001; 0.089)	0.033 (−0.045; 0.112)	0.079 (0.035; 0.123)	0.064 (0.001; 0.126)	0.001 (−0.036; 0.039)	0.001 (−0.005; 0.008)	0.974 (0.896; 1.06)
<LOD *n* = 839 ≥0.069 *n* = 75	Ref. −0.319 (−0.764; 0.126)	Ref. −0.057 (−0.406; 0.291)	Ref −0.236 (−0.82; 0.344)	Ref. 0.431 (0.151; 0.712)	Ref. −0.104 (−0.524; 0.32)	Ref. −0.259 (−0.513; −0.005)	Ref. 0.0325 (−0.015; 0.080)	Ref. 0.525 (0.256; 1.076)
Log-Mercury *n* = 914	0.030 (−0.016; 0.076)	0.017 (−0.022; 0.056)	0.051 (−0.011; 0.11)	0.050 (0.008; 0.092)	0.056 (−0.001; 0.11)	0.027 (−0.008; 0.06)	0.002 (−0.004; 0.007)	0.963 (0.889; 1.043)
<LOD *n* = 837 ≥0.079 *n* = 77	Ref. −0.060 (−0.390; 0.269)	Ref. −0.012 (−0.271; 0.248)	Ref. 0.371 (−0.093; 0.83)	Ref. 0.531 (0.254; 0.809)	Ref. 0.522 (0.147; 0.897)	Ref. 0.008 (−0.26; 0.274)	Ref. −0.014 (0.054; 0.027)	Ref. 0.603 (0.307; 1.184)
Log-Copper/Zinc Ratio *n* = 914	0.286 (0.146; 0.425)	0.175 (0.061; 0.288)	0.450 (0.250; 0.649)	0.153 (0.037; 0.268)	0.325 (0.170; 0.481)	0.245 (0.142; 0.347)	−0.018 (−0.03; −0.002)	1.420 (1.160; 1.740)

Abbreviations: NGAL, neutrophil gelatinase-associated Lipocalin; KIM-1, Kidney Injury Molecule 1; α-1MG, α-1-microglobulin; OPN, Osteopontin; Cys-C, Cystatin C; CLU, Clusterin; LOD, limit of detection; eGFR, estimated glomerular filtration rate (miL/min/1.73 m^2^); ACR, albumin–creatinine ratio (mg/g-creat). Models adjusted for age (years), sex (female/male), body mass index (normal/underweight/overweight/obese), poverty (yes/no). Early kidney damage biomarkers and metals were adjusted for urinary density and expressed as ng/mL. Cadmium <LOD, *n* = 839, ≥0.069, *n* = 75; Mercury <LOD, *n* = 837, ≥0.079, *n* = 77.

## Data Availability

The datasets used and analyzed during the current study are available from the corresponding author upon reasonable request.

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
