# Peer review of "Relationship Between Urinary Copper, Zinc, and Cadmium and Kidney Damage Biomarkers in Young People"

_ijms, 2025, doi:10.3390/ijms26167980_

Round 1
Reviewer 1 Report
Comments and Suggestions for Authors
The article is about a significant public health problem related with Cd associated nephrotoxicity however, there are some limitations that should be corrected before publication:
- the problem and the objectives of the study are not clearly described.
- there is a lot of information related to heavy metals and nephrotoxicity carried out in Mexico, which are not taken into account neither in the introduction nor in the discussion.
- many of the references are not written or discussed in the proper context, as an example reference 41
- in the material and methods section, describe in a better way how the study population was chosen, whether it was a school population, a particular small town, an open population, etc., and why that particular population was chosen. Describe the general characteristics surrounding the study population and what risk factors have been associated in this population to carry out the sampling in them.
- How do the authors explain that 91.8% of Cd and 94.3% of Hg determination in urine below the detection limit.
- explain much better how the authors carried out the association studies if they were able to determine Cd U values in only 10% of the population studied. Would they have the same results if they performed the analyses only on the samples in which they were able to determine the Cd?
- If CdU and HgU were detected in only 10% of the samples, does this represent a public health problem? Explain this further in the discussion.
- Table number 1 clarifies the demographic and laboratorial characteristics of the group of patients, since in some cases they handle logarithms and in other absolute values there is no uniformity in the article.
- It would be preferable if they could make a table comparing people with CdU undetectable vs. detectable CdU and evaluate if there were differences in kidney injury markers. The same could be done comparing groups of patients with high Cu/Zn ratio vs. those with low ratio and kidney injury markers.
improving the descriptive quality of the discussion
Author Response
Please see attached document with detailed responses
The article is about a significant public health problem related with Cd associated nephrotoxicity however, there are some limitations that should be corrected before publication:
- The problem and the objectives of the study are not clearly described.
“This study aims to evaluate the impact of the Zn/Cu-R on kidney damage biomarkers in young individuals exposed to low-level Cd, considering its potential role in kidney health.”.
The objective of this study was to evaluate the Cu/Zn ratio as a kidney damage biomarker and identify if it participates as a protector or an enhancer of damage when the pediatric population is exposed to toxic components, in this particular case to Cadmium.
- There is a lot of information related to heavy metals and nephrotoxicity carried out in Mexico, which are not taken into account neither in the introduction nor in the discussion.
R= Done. The introduction was added with more information about the nephrotoxic elements and their situation in Mexico
- Many of the references are not written or discussed in the proper context, as an example reference 41.
R= Thanks for your comments. We made a few changes to the text, intending to put some of the references in a better context or add other references that could make more sense.
- In the material and methods section, describe in a better way how the study population was chosen, whether it was a school population, a particular small town, an open population, etc., and why that particular population was chosen. Describe the general characteristics surrounding the study population and what risk factors have been associated in this population to carry out the sampling in them.
R= In the study design section, the points solicited by the review were added, including backgrounds over CKD in the population selected.
References
- Ortega-Romero, M., P. Mendez-Hernandez, M. D. C. Cruz-Angulo, A. M. Hernandez-Sanchez, A. C. Alvarez-Elias, R. Munoz-Arizpe, F. Sales-Heredia, G. Aguilar-Madrid, C. A. Juarez-Perez, V. Soto, T. Valades, N. Olvera-Rivas, G. T. Obrador-Vera, O. C. Barbier, and M. Medeiros. 'Chronic Kidney Disease in Children Aged 6-15 Years and Associated Risk Factors in Apizaco, Tlaxcala, Mexico, a Pilot Study.
- How do the authors explain that 91.8% of Cd and 94.3% of Hg determination in urine below the detection limit.
R= The study population represents an apparently healthy pediatric population. It is not a population with occupational exposure to contaminants, nor is it a population with a known source of exposure. Existing information indicates that the study area has a high risk of contamination; however, the contaminants present include pesticides, volatile organic compounds, anilines, etc. (1, 2), our working group chose to study exposure to metals and metalloids. Regarding the question raised by the reviewer, Cd and Hg are elements considered toxic and are most often the result of contamination by human activities.
We attribute the presence of Cu and Hg in less than 10% of participants due to:
- First, soil studies have shown that Cd, for example, is absent or present in low concentrations in the area where the study was conducted (3). Regarding water as a source of contamination, reports indicate that the levels observed are outside quality standards (4). However, our population reported that their water consumption, whether for drinking or cooking, came primarily from bottled water or water treatment plants. This may explain why most participants are not exposed to these elements.
- Secondly, the industrial activities present in the area correspond to the textile industry, related to chemical pollutants, dyes and anilines, as contaminants. Although the automotive industry is also present, related to metals and metalloids, official reports indicate that its waste is mainly discharged into the Zahuapan River, which rises in the state of Tlaxcala and flows into the state of Puebla, causing the accumulation of pollutants (Cd and Hg) to increase as the river advances, causing problems in Puebla and not at the source of the river (Tlaxcala). Furthermore, the Zahuapan River has no use for the population studied; it is not used for any agricultural or domestic activity, and the population is aware of the environmental problems of this body of water, so they usually take measures and avoid it (1 and 2).
These factors could explain why Cd and Hg only were present in 10% of participants. However, it should be noted that both elements are known to be nephrotoxic, so their presence in the participants, even at low concentrations and in less than 10% of the participants, is significant and makes it necessary to study the consequences of their presence in the pediatric population.
References
- Gobierno de México. Plan integral de atención a emergencia sanitaria y ambiental de la cuenca del río Alto Atoyac. CONAHCYT. 2023.
- Gobierno de Tlaxcala. Datos sobre la situación ambiental y de salud del estado de Tlaxcala y la Cuenca de Atoyac-Zahuapan. México; 2023.
- Calzada Mendoza, Jacqueline Mireya. (2007). "Mapas geoquímicos de metales pesados de suelos del Estado de Tlaxcala, México". (Tesis de Maestría). UNAM, México. Recuperado de https://repositorio.unam.mx/contenidos/97244
- Explain much better how the authors carried out the association studies if they were able to determine Cd U values in only 10% of the population studied.
R = Exposure biomarkers with values below the limit of detection (LOD) are equivalent to left-censored data because we do not know the exact value. This data is more informative than missing data and is considered missing not at random (MNAR) (Domthong et al., 2014). Although we do not know the exact CdU values, these subjects are in the group with the lowest exposure levels in the study population. Ignoring these data in statistical analysis is inappropriate because the results are biased. For this reason, we evaluated the possibilities for appropriate management of these data and chose two approaches:
- The first approach was a dichotomous classification of exposure performed for subjects who had undetectable CdU values (<LOD) and those who had detectable values (>=0.069), which is a practical way to manage data when the percentage of values below the limit of detection is high. It is considered a conservative nonparametric method because it does not assume any behavior in the data; it uses percentiles or ranges of the data set and will produce more reliable results. If this is the outcome, we can use logistic regression to evaluate the association with independent variables. The median or mean differences are adequate in bivariate analysis (Helsel, 2011).
- The second approach was the imputation of left-censored data using the iterative Markov Chain Monte Carlo (MCMC) data augmentation algorithm, which consists of two steps: 1) assigning a value using information from the covariance structure under the assumption of a left-truncated distribution and 2) estimating the mean and variance parameters from the data from the first point. The sequence of these two steps generates a Markov chain, which, after multiple iterations, yields adequate values for observations below the LOD. This method conserves the covariance matrix structure and is adequate for multiple biomarkers (Palarea-Albaladejo & Martín-Fernández, 2015).
6.1. Would they have the same results if they performed the analyses only on the samples in which they were able to determine the Cd?
R= No, because excluding subjects below the detection limit would give us biased estimates of the degree of exposure, the effect of this exposure, and its association with biomarkers of kidney damage. The overestimation of the effect, loss of power, and increase in the possibility of an error type 2 are possible effects described in the literature (Herbers et al., 2021). We made an additional statistical analysis, restricting observations to =>LOD for cadmium (n=75) and mercury (n=77) and evaluated the association with early kidney damage biomarkers. In this analysis observed the following phenomena: 1) the coefficients ß and OR are highest in the restricted analysis compared with the result of the total sample, for example in the association of log-mercury with log-a-1MG, in the complete sample (n=914) the coefficient is [(ß=0.051; CI 95% (-0.011; 0.113)] and in the in the restricted sample (n=77) the coefficient is [(ß=0.324; CI 95% (-0.080; 0.728)]. 2) In some cases, the association changed the direction, for example, log-Cd associated to log-KIM-1 in the complete sample (n=914) [(ß=0.044; CI 95% (-0.001; 0.089)] vs in the restricted sample (n=75) the [(ß=-0.169; CI 95% (-0.587; 0.246)]. 3) Observed a loss of statistical significance, for example in the association of log-mercury with log-OPN, in the complete sample (n=914) [(ß=0.050; CI 95% (0.008; 0.092] vs in the restricted sample (n=77) [(ß=-0.205; CI 95% (-0.058; 0.468)]; 4) The association changed the direction and observed a loss of statistical significance, for example in the association of log-cadmium with log-OPN, in the complete sample (n=914) [(ß=0.123; CI 95% (0.079; 0.035] vs the restricted sample (n=75) [(ß=-0.067; CI 95% (-0.396; 0.261)]. The complete information is included in Table 1 (attached file). Due to these effects, we considered it inappropriate to restrict the analysis only to detectable values.
References
Domthong, U., Parikh, C. R., Kimmel, P. L., & Chinchilli, V. M. (2014). Assessing the agreement of biomarker data in the presence of left-censoring. BMC Nephrology, 15(1), 144. https://doi.org/10.1186/1471-2369-15-144
Helsel, D. R. (2011). Statistics for Censored Environmental Data Using Minitab and R. Wiley. https://books.google.com.mx/books?id=cgez4u8bTpoC
Herbers, J., Miller, R., Walther, A., Schindler, L., Schmidt, K., Gao, W., & Rupprecht, F. (2021). How to deal with non-detectable and outlying values in biomarker research: Best practices and recommendations for univariate imputation approaches. Comprehensive Psychoneuroendocrinology, 7, 100052. https://doi.org/10.1016/j.cpnec.2021.100052
Palarea-Albaladejo, J., & Martín-Fernández, J. A. (2015). zCompositions—R package for multivariate imputation of left-censored data under a compositional approach. Chemometrics and Intelligent Laboratory Systems, 143, 85–96. https://doi.org/10.1016/j.chemolab.2015.02.019
- If CdU and HgU were detected in only 10% of the samples, does this represent a public health problem? Explain this further in the discussion.
R= No, we could not consider it such a public health problem, for this to be possible the percentage of individuals exposed to these elements should be higher or the concentrations found should exceed the values considered as maximum permissible limits (1, 2). In this work what we try to show is that the presence of toxic elements such as Cd can cause alterations in kidney health when these toxic elements are in the body at the same time that we observe anomalies in essential elements for the body, indicating that the interactions or synergies between the elements can cause effects that could be deleterious to the body. Therefore, no, we could not consider the concentrations or percentages of Cd and Hg observed in the population as a public health problem, but what is interesting in this approach is the capacity for damage or alterations to the body when these elements interact with others, even at concentrations that are considered safe.
References
- Kuo, P.F., et al., Association of low-level heavy metal exposure with risk of chronic kidney disease and long-term mortality. PLoS One, 2024. 19(12): p. e0315688.
- Smereczanski, N.M. and M.M. Brzoska, Current Levels of Environmental Exposure to Cadmium in Industrialized Countries as a Risk Factor for Kidney Damage in the General Population: A Comprehensive Review of Available Data. Int J Mol Sci, 2023. 24(9).
- Table number 1 clarifies the demographic and laboratorial characteristics of the group of patients, since in some cases they handle logarithms and in other absolute values there is no uniformity in the article.
R= Thanks for your comments. We included the logarithms description. However, we consider it essential to conserve the absolute value of the biomarkers and metals to be more informative concerning urinary metal and biomarker levels.
- It would be preferable if they could make a table comparing people with CdU undetectable vs. detectable CdU and evaluate if there were differences in kidney injury markers. The same could be done comparing groups of patients with high Cu/Zn ratio vs. those with low ratio and kidney injury markers.
R= Done, we included a bivariate analysis to compare groups according to metal levels.
Comments on the Quality of English Language, improving the descriptive quality of the discussion
We thank the comments that improved our manuscript

Reviewer 2 Report
Comments and Suggestions for Authors
The authors conducted an observational study to assess kidney damage biomarkers (OPN, KIM-1, CLU, NGAL, and Cys-C) using Luminex MAGPIX and trace metal levels (Cd, Hg, Cu, Zn) using ICP-mass spectrometry in urine samples from 914 adolescents (young individuals) aged 11–18 to explore the correlation between renal parameters and metal exposure.
Here are some of my suggestions:
1.
In Table 3, "Urinary metals excretion according to urinary cadmium levels (≥0.069, n=? vs. <LOD [limit of detection], ref, n=?)," the sample size for each group should be specified.
2.
In Table 4, the sample size for each group should also be specified in the footnote for both urinary cadmium levels (≥0.069, n=? vs. <LOD, ref, n=?) and mercury levels (≥0.079, n=? vs. <LOD, ref, n=?).
4.
In Table 3, urinary metals excretion is presented according to urinary cadmium levels (categorized as ≥0.069 and <LOD [limit of detection]). A decrease in the log-Cu/Zn ratio (β: -0.237, 95% CI: -0.352, -0.123) was observed in the ≥0.069 category compared to the <LOD category.
In Table 4, the log-Cu/Zn ratio showed a significant positive association with all biomarkers of early kidney damage, including Log-NGAL, Log-KIM-1, Log-α-1MG, Log-OPN, Log-Cys-C, and CLU (β ≥ 0.153, 95% CI does not include zero, indicating statistical significance). Additionally, the log-Cu/Zn ratio was negatively associated with eGFR (β: -0.018, 95% CI: -0.033, -0.002) and was linked to an increased risk of albuminuria (OR≥30 vs. <30: 1.42, 95% CI: 1.16, 1.74).
However, it is recommended to analyze the interaction or effect modification between urinary cadmium levels and the log-Cu/Zn ratio in relation to early kidney damage, eGFR, and the risk of albuminuria.
4.
The limit of detection (LOD) refers to the lowest concentration of urinary metal excretion that can be reliably detected using a given analytical method. For example, in the case of cadmium, values above the LOD (e.g., cadmium ≥0.069 ng/mL) are considered detectable. However, if certain values significantly exceed the LOD, they may fall outside the expected detectable range or represent extreme values (outliers).
In such cases, do these values indicate outliers beyond the expected range?
If so, could they be considered unreliable data?
If such extreme values exist, would they introduce bias into the statistical analysis?
Author Response
Please see attached file
We thank the reviewers comments

Round 2
Reviewer 1 Report
Comments and Suggestions for Authors
ALL CORRECTIONS WERE MADE CORRECTLY
Author Response
Thank you

Reviewer 2 Report
Comments and Suggestions for Authors
None
Author Response
Thank you
